# Giant room temperature anomalous Hall effect and tunable topology in a ferromagnetic topological semimetal $Co_2MnAl$

Peigang Li[1,6], Jahyun Koo[2,6], Wei Ning[3✉], Jinguo Li[4], Leixin Miao[5], Lujin Min[3,5], Yanglin Zhu[1,3], Yu Wang[1,3], Nasim Alem[5], Chao-Xing Liu[3], Zhiqiang Mao[1,3✉] & Binghai Yan[2✉]

Weyl semimetals exhibit unusual surface states and anomalous transport phenomena. It is hard to manipulate the band structure topology of specific Weyl materials. Topological transport phenomena usually appear at very low temperatures, which sets challenges for applications. In this work, we demonstrate the band topology modification via a weak magnetic field in a ferromagnetic Weyl semimetal candidate, $Co_2MnAl$, at room temperature. We observe a tunable, giant anomalous Hall effect (AHE) induced by the transition involving Weyl points and nodal rings. The AHE conductivity is as large as that of a 3D quantum AHE, with the Hall angle ($\Theta^H$) reaching a record value ($\tan \Theta^H = 0.21$) at the room temperature among magnetic conductors. Furthermore, we propose a material recipe to generate large AHE by gaping nodal rings without requiring Weyl points. Our work reveals an intrinsically magnetic platform to explore the interplay between magnetic dynamics and topological physics for developing spintronic devices.

[1] Department of Physics and Engineering Physics, Tulane University, New Orleans, LA 70118, USA. [2] Department of Condensed Matter Physics, Weizmann Institute of Science, Rehovot 7610001, Israel. [3] Department of Physics, Pennsylvania State University, University Park, State College, PA 16802, USA. [4] Superalloys Division, Institute of Metal Reseach, Chinese Academy of Sciences, 110016 Shenyang, China. [5] Department of Materials Science and Engineering, Pennsylvania State University, University Park, State College, PA 16802, USA. [6]These authors contributed equally: Peigang Li, Jahyun Koo. ✉email: wvn5038@psu.edu; zim1@psu.edu; binghai.yan@weizmann.ac.il

The Weyl semimetal (WSM)[1–7] is characterized by the linear band-crossing points, called Weyl points, which exhibits monopole-type structure of the Berry curvature[8,9], leading to many exotic properties, such as the Fermi arc surface states[1], the chiral anomaly effect[10,11], and the anomalous Hall effect (AHE)[12,13]. This topological phase has been discovered in materials such as TaAs[14–18] and $MoTe_2$[19–22] recently. In these materials, once the crystal is formed, their positions and energies of Weyl points are usually predetermined by material parameters, including the crystal structure and spin–orbit coupling (SOC), and can hardly be manipulated freely.

A Weyl point is robust in the sense it does not require any symmetry protection except the lattice translation. Inside a mirror plane that prohibits the Berry-curvature monopole, the Weyl point disappears, while 1D nodal lines may emerge due to the mirror symmetry[23–25]. The nodal line displays a $\pi$ Zak–Berry phase accumulated along a loop circling the nodal line, which induces Shockley-like[26,27] surface states (called drum-head surface states[28]). However, the nodal line exhibits zero Berry curvature in its vicinity and thus does not generate an AHE. If the mirror symmetry is broken, the nodal line gets gapped out and evolves into a pair of Weyl points (e.g., ref. [29]). The topology change can be directly probed by the AHE or observed in other topology-induced phenomena. Therefore, the rotation of the magnetization orientation, which sensitively switches the mirror symmetry in a ferromagnet, provides a powerful tool to tune the topological band structure.

Although there have been several reported/predicted magnetic WSMs, few of them are appropriate for tuning the band structure topology via magnetic field. For the antiferromagnetic WSMs $Mn_3Sn$ and its sister compound $Mn_3Ge$[30–34], in spite of their AHE varying with the rotation of spins within the spin easy plane, it remains unclear how their Weyl points and nodal lines (if they exist) evolve because of the complexity in the band structure. $Co_3Sn_2S_2$ was reported to be a layered ferromagnetic (FM) WSM[35–39], which presents only Weyl points near the Fermi energy. Since its magnetization favors only the out-of-plane direction with the Curie temperature at 177 K, the band tuning by magnetization is challenging, which is similar to the case of a recently reported FM nodal line semimetal $Fe_3GeTe_2$[40]. Recently, Co-based Heusler alloy compounds $Co_2XZ$ (X=V, Zr, Nb, Ti, Mn, Hf; Z=Si, Ge, Sn, Ga and Al), previously known as half metallic ferromagnets, were predicted to host FM WSM phases[41–47]. Since many of these materials are soft ferromagnets with their Curie temperatures far above room temperature, a weak external magnetic field can easily drive their magnetization to rotate in a wide temperature range. Hence, these materials provide ideal platforms to tune the band structure topology.

Although $Co_2XZ$ allows for many different element combinations, experimental studies on their possible exotic properties induced by the expected WSM states are sparse, which is possibly due to the difficulty of the single crystal growth of this family of materials. Up-to-date, $Co_2MnGa$ is the only member that was reported to have distinct properties associated with the FM WSM state, that is, a giant AHE and anomalous Nernst effect[48,49]. The band structure calculations show that $Co_2MnGa$ as well as other $Co_2XZ$ members have both Weyl points and nodal rings, and the giant AHE in these materials is generally believed to originate from Weyl points[42,43,46,48,50], with the anomalous Hall conductivity (AHC) being proportional to the separation of Weyl points. In theory, the largest possible Weyl point separation is the Brillouin size, which should induce the quantized AHC, that is, a 3D quantum AHE (QAHE). How can a magnetic WSM reach the 3D QAHE or giant AHE? What is the recipe to find such materials? Answers to these questions are not only of fundamental importance, but also likely leads to technological applications.

In this study, we give clues to these questions through the study of AHE of a Heusler alloy $Co_2MnAl$. This compound has attracted much attention due to its potential applications in spintronics[51] and Hall sensors[52,53], as well as its recent prediction of FM WSM[42]. $Co_2MnAl$ possesses several structural phases with different disorder types, including $L2_1$ (Mn, Al ordered), B2 (Mn, Al disordered), $DO_3$ (Co, Mn disordered), A2 (Co, Mn, Al disordered)[52], and only the $L2_1$ phase with the cubic space group of $Fm\bar{3}m$ was predicted to host the FM WSM phase. However, synthesis of the $L2_1$ phase has been challenging. All previous studies on this material were based on either polycrystalline or thin film samples with the B2 phase. Our recent success in growing $Co_2MnAl$ single crystals with the $L2_1$ phase has enabled us to observe its fascinating properties originating from the band topology. We found that this material exhibits very large AHC, up to $1300\,\Omega^{-1}\,cm^{-1}$ at room temperature; more noticeably, its room temperature anomalous Hall angle (AHA) $\Theta^H$ reaches a new record value among all magnetic conductors, with $\tan\Theta^H = 0.21$. Furthermore, its AHE can be tuned by rotation of the magnetization axis as expected. From the comparison between our experimental results and theoretical calculations, we found that the gapped nodal rings induce large Berry curvature, which is responsible for the observed giant AHE. The consistency between our experiments and theory suggests a new mechanism of creating a large AHE by nodal ring's gap opening; through this mechanism, an AHC as large as that of a 3D QAHE can be reached. This mechanism is distinct from the previous understanding of the AHE originating from Weyl points for $Co_2MnGa$ and similar materials.

## Results

**Characterization of single crystals.** The $Co_2MnAl$ single crystals used in this study were grown using floating-zone (FZ) technique (see the "Methods" section). Figure 1b shows an optical image of typical crystals. As shown in Fig. 1a, in the $L2_1$ phase, Mn and Al are ordered, which is manifested by the (111) diffraction spot/peak in an electron/X-ray diffraction pattern according to previous studies[54]. From our electron diffraction analyses, we have indeed observed the (111) diffraction spot in the electron diffraction pattern taken along the [110] zone axis (Fig. 1d), indicating that our $Co_2MnAl$ crystals surely have the $L2_1$ structure phase. This is further corroborated by scanning transmission electron microscopy (STEM) imaging shown in Fig. 1c, where the periodic, alternating distribution of Mn and Al (left inset to Fig. 1b) can be seen clearly from the atomic intensity line due to Z-contrast. Furthermore, we have also used the high-angle annular dark-field (HAADF) STEM technology to check the sample homogeneity, which confirmed that uniform $L2_1$ phase is formed throughout the entire sample.

The magnetic properties of $Co_2MnAl$ single crystals were characterized through magnetization $M$ measurements. Figure 1e shows the isothermal magnetization data at 2 and 400 , measured with the magnetic field applied parallel and perpendicular to the (001)/(111) plane, respectively. Field sweep measurements of $M$ reveal small magnetic hysteresis, and the remnant magnetization is ~0.1 and $0.2\mu_B$ per formula unit, respectively, for $B\perp$ (001) and $B\parallel$ (111) at 2 and 400 K (see the insets to Fig. 2e). These data are consistent with the previous report that $Co_2MnAl$ is a soft ferromagnet, with the saturation moment $M_s$ of ~$4.2\pm0.2\mu_B$ per formula[54]. The small decrease of $M_s$ from 2 to 400 K suggests that its Curie temperature is far above room temperature, which cannot be probed in our superconducting quantum interference device (SQUID) magnetometer. The previously reported $T_c$ for this material is 726 K[54]. Moreover, our magnetization data also reveal its FM properties are anisotropic and the (111) plane is the easy spin plane.

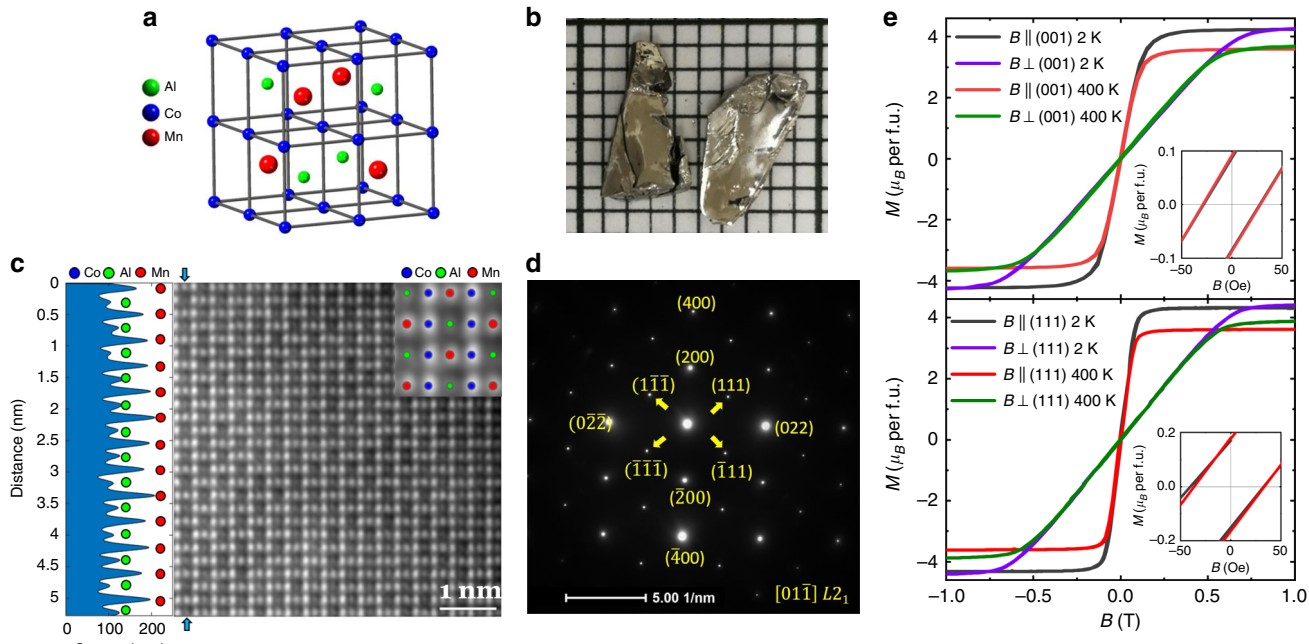

**Fig. 1 Structure and magnetization of Co₂MnAl single crystals. a** Schematic crystal structure of L2₁-type Co₂MnAl; **b** the optical graph of Co₂MnAl single crystals grown by the optical floating-zone method; **c** HAADF-STEM image of a selected area taken along the [110] zone axis. The left panel shows alternating intensity due to Z-contrast, revealing the ordering of Mn and Al; the right inset presents a magnified image with the atoms overlaid on top. **d** The selected area diffraction pattern taken along the [110] zone axis. **e** The isothermal magnetization data at 2 and 400 K, measured with the magnetic field applied parallel and perpendicular to the (001)/(111) plane, respectively. The insets show the enlargement of the magnetization around zero field.

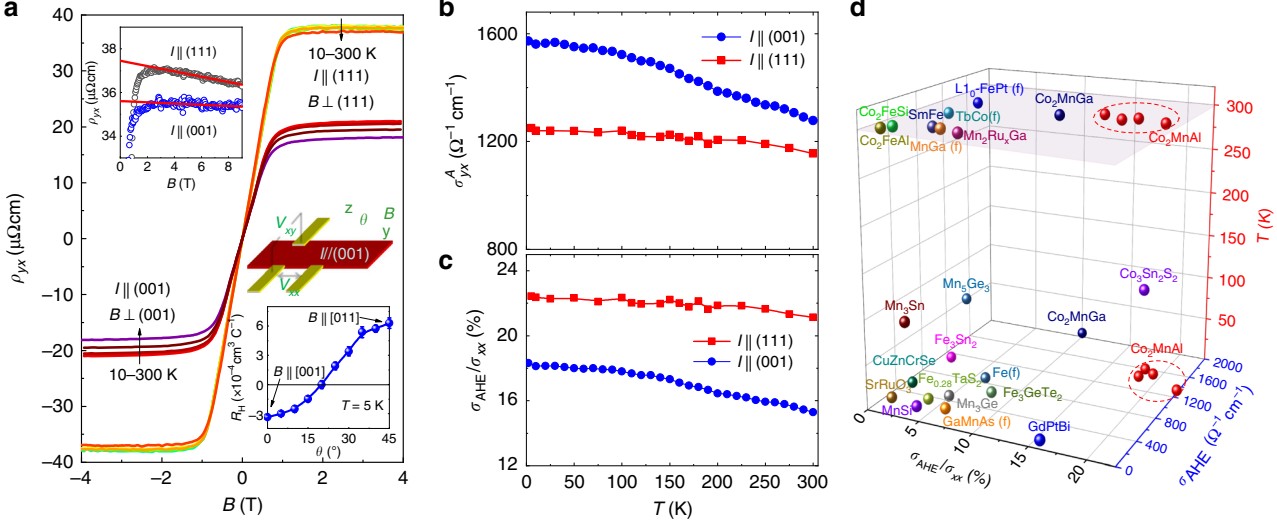

**Fig. 2 The anomalous Hall effect measurement. a** Magnetic field-dependent Hall resistivity ($\rho_{xy}$) at various temperatures for two Co₂MnAl samples with different orientations. One sample was measured with $I \parallel (001)$ plane and $B \perp (001)$ and the other with $I \parallel (111)$ and $B \perp (111)$. Left inset: $\rho_{yx}$ vs. $B$ at 300 K, with the data for $I \parallel (001)$ being offset for clarity. Bottom right inset, Hall coefficient $R_H$ as a function of the orientation angle of magnetization for the sample with $I \parallel (001)$. The middle right inset shows the experimental setup for the angular-dependent Hall effect measurements. **b** Temperature dependence of anomalous Hall conductivity (AHC) ($\sigma_{yx}^A$) for the samples shown in **a**. **c** The temperature dependence of $\tan \Theta^H = \sigma_{yx}^A/\sigma_{xx}$ at 2 T [$\Theta^H$, anomalous Hall angle (AHA)]. **d** Comparison of AHC and $\tan \Theta^H$ between Co₂MnAl and other magnetic conductors; "f" refers to thin film materials. Co₂MnAl exhibits both large AHA and large AHC in a wide temperature range. In addition to the data collected on the samples shown in **a**, we have also included the data measured on another two additional Co₂MnAl samples with $I \parallel (001)$ and $I \parallel (010)$ in this figure. The AHC and AHA data of other cited magnetic materials and the related references are given in Supplementary Table 1.

**AHE measurement.** We have performed Hall resistivity $\rho_{xy}$ measurements on the Co₂MnAl samples with the electric current applied to the (001), (010), and (111) planes, respectively. Figure 2a presents the field dependence of $\rho_{xy}$ in the 10–300 K temperature ranges for two typical samples with $I \parallel (001)$ and $I \parallel$ (111). In general, $\rho_{xy}$ of an FM conductor can be expressed as $\rho_{xy} = R_H B + 4\pi R_s M$, where $R_H B$ represents normal Hall contribution due to Lorentz effect ($R_H$, Hall coefficient) and $4\pi R_s M$ is anomalous Hall contribution $\rho_{xy}^A$. As seen in Fig. 2a, $\rho_{xy}$ shows a similar magnetic field dependence as the isothermal

magnetization shown in Fig. 1e, indicating that the anomalous Hall contribution (proportional to $M$) is dominant and the normal contribution is very small and estimated to be 0.3–3% of $\rho_{xy}$ in the 2–9 T field range, where $M$ is saturated. $\rho_{xy}^A$ at room temperature shows large values, ~17.9 μΩcm for $I \parallel (001)$ and 36.9 μΩcm for $I \parallel (111)$. We have also measured field dependence of longitudinal resistivity $\rho_{xx}$ at various temperatures for these samples; the representative data from these measurements are presented in Supplementary Fig. 5. From the $\rho_{xy}^A$ obtained after subtracting normal Hall contribution and the $\rho_{xx}$ data, we have derived the AHC $\sigma_{yx}^A$ as a function of temperature at 2 T for the $I \parallel (001)$ and $I \parallel (111)$ samples (Fig. 2b) via tensor conversion, that is, $\sigma_{yx}^A = \rho_{xy}^A / (\rho_{xy}^{A\,2} + \rho_{xx}^2)$. Both samples are found to show exceptionally large AHC. At 2 K, $\sigma_{yx}^A$ reaches ~1600 $\Omega^{-1}$ cm$^{-1}$ for the $I \parallel (001)$ sample. Upon increasing temperature, the AHC decreases slightly and remains up to ~1300 $\Omega^{-1}$ cm$^{-1}$ at room temperature. For the $I \parallel (111)$ sample, the AHC is almost temperature independent and just decreases from ~1250 $\Omega^{-1}$ cm$^{-1}$ at 2 K to ~1190 $\Omega^{-1}$ cm$^{-1}$ at room temperature. Such a large AHC at room temperature in Co$_2$MnAl has never been reported due to the lack of bulk single crystals. The previously reported Hall effect measurements on Co$_2$MnAl were performed on thin film samples with the B2 phase[52,53]. The giant room temperature AHC revealed in our experiment, as well as its tunability by a weak magnetic field as discussed below, suggests that Co$_2$MnAl is a promising material for device applications such as Hall sensor.

Besides large AHC, Co$_2$MnAl also shows a large AHA, $\Theta^H$ (tan $\Theta^H = \sigma_{yx}^A / \sigma_{xx}$), as shown in Fig. 2c. For the $I \parallel (111)$ sample, its tan $\Theta^H$ value is as large as 21% even at room temperature, which is a record value among either trivial or topological magnetic conductors, as far as we know. This can be seen in Fig. 2d, where we have compared AHC and AHA between our Co$_2$MnAl samples and those magnetic materials known as having large $\sigma_{yx}^A$ and $\Theta^H$. Magnetic WSMs are generally expected to have large AHC and AHA due to the Berry curvature induced by non-trivial band topology. Among reported magnetic WSMs, while some of them indeed exhibit large AHC and AHA, they occur mostly at low temperatures. For instance, the tan $\Theta^H$ of the FM WSM Co$_3$Sn$_2$S$_2$ can also reach 20%, but it can be observed only below 120 K[35]. The magnetic field induced WSM GdPtBi was also reported to have large AHA, with the largest tan $\Theta^H$ value of ~10% being probed only below 10 K[55]. As noted above, Heusler compound Co$_2$MnGa has also been reported to have very large AHE[48]; its AHC reaches ~2000 $\Omega^{-1}$ cm$^{-1}$ at 2 K, but decreases down to 1000 $\Omega^{-1}$ cm$^{-1}$ at room temperature. Its room temperature tan $\Theta^H$ is 12%, about half of the largest value we observed in Co$_2$MnAl. Compared to Co$_2$MnGa, Co$_2$MnAl shows a much weaker temperature dependence in AHC and AHA for $I \parallel (111)$ (Fig. 2b, c), which is important for device applications.

In general, an AHE may either originate from extrinsic mechanism (i.e., skew scattering and side jump), or arises from intrinsic Berry-curvature contribution[9], or combined extrinsic and intrinsic contributions[8]. From the comparison of the temperature dependence of longitudinal resistivity $\rho_{xx}(T)$ and AHC $\sigma_{yx}^A(T)$ (Fig. 2c), we find large AHC revealed in our experiments for Co$_2$MnAl is of intrinsic origin. In Supplementary Fig. 6, we show the $\rho_{xx}(T)$ data for both $I \parallel (001)$ and $I \parallel (111)$, from which we observe clear temperature dependence as well as localization behavior below 40 K, indicating the scattering rate varies with temperature. In contrast, $\sigma_{yx}^A$ remains nearly unchanged from several K to 300 K for $I \parallel (111)$ despite the scattering rate variation, which is a proof of the intrinsic nature of the AHE according to the theory of AHE in ferromagnets[8,9].

Furthermore, our theoretical studies presented below also show the intrinsic origin of AHE in Co$_2$MnAl. The $\sigma_{yx}^A$ calculated based on Berry curvature (Supplementary Fig. 1) is close to the experimental value noted above. Additionally, our theory not only explains why $\sigma_{yx}^A(T)$ for $I \parallel (001)$ has a slightly negative slope as compared to the $I \parallel (111)$ case as shown in Fig. 2c, but also predicts $\sigma_{yx}^A$ can be tuned by the rotation of magnetization, which is also consistent with our experiments, as to be discussed below.

## Discussion

To understand the intrinsic AHE of Co$_2$MnAl, we have preformed theoretical calculations on the band structure and AHC of Co$_2$MnAl. The intrinsic AHE originates from the band structure and is robust against disorders and defects. The magnetization direction can modify the AHE, because it determines the symmetry and consequently the band structure of the system. We show the calculated AHC ($\sigma_{yx}^A$) in Supplementary Fig. 1. It is not surprising to find that the magnetization along [001] and [111] leads to slightly different amplitude in $\sigma_{yx}^A$. Near the charge neutral point, the [111] value is slightly lower than the [001] value and the [111] AHC shows weaker temperature dependence than the [001] case, which is semi-quantitatively consistent with our experiment and is attributed to the band structure anisotropy. $\sigma_{yx}^A$ exhibits different dependence on the chemical potential between the [001] and [111] cases, leading to different temperature dependence of $\sigma_{yx}^A$ as shown in the calculation (Supplementary Fig. 1b) and experiment (Fig. 2b).

We examine the origin of the strong AHE in the band structure by taking the [001] magnetization for example. In an energy window of $-0.2$ to $+0.1$ eV with respect to the charge neutral point ($\mu = 0$), we find four nodal rings centered at the $Z$ point of the FCC Brillouin zone (see more information in the Supplementary information). We note that these nodal rings are protected by the mirror symmetry of the $k_z = 0$ ($k_z = 0$ and $\pi$ planes are equivalent in the FCC Brillouin zone), as shown in Fig. 3c and Supplementary Fig. 4b. However, corresponding nodal rings at the $k_x = 0$ and $k_y = 0$ planes are gapped out because of the mirror symmetry breaking caused by the [001] magnetic moment (as shown in Supplementary Fig. 2c). These four nodal rings, which are denoted by #1–4 in the following discussions, are induced by band crossing between the highest two valence bands and lowest two conduction bands (see Fig. 3c). In a previous study (ref. [44]), only nodal rings #2 and #3 were investigated in a similar compound Co$_2$MnGa without considering the SOC. Without SOC, the magnetization axis does not pick up a specific direction. Thus, these nodal rings appear in all mirror planes, that is, $k_{x,y,z} = 0$. As shown in Fig. 3b, four nodal rings in different planes interconnect each other to form topological Hopf nodal links[44,56]. The SOC leads the magnetic moment to couple to the lattice, thus reducing the symmetry. The preserved nodal rings in the mirror plane do not contribute any Berry curvature (see Supplementary Fig. 2), which is characterized by a $\pi$ Berry phase accumulated along a loop interlocking the nodal ring. Thus, these four nodal rings do not contribute to the AHE. In contrast, the gapped nodal rings generate large Berry curvature and thus give rise to the huge AHC (see Supplementary Fig. 2) as observed in our experiments. Four peaks of $\sigma_{yx}^A$ in this energy window are dominantly contributed by these four gapped nodal rings (see Supplementary Fig. 3). Near the charge neutral point, AHC exhibits a large value because of the existing peak at $\mu = 25$ meV, which is mainly induced by the gapped nodal ring #3 (see Supplementary Fig. 3), rather than Weyl points. It should be clarified that although Co$_2$MnAl also hosts normal carriers (see Supplementary Fig. 4) besides

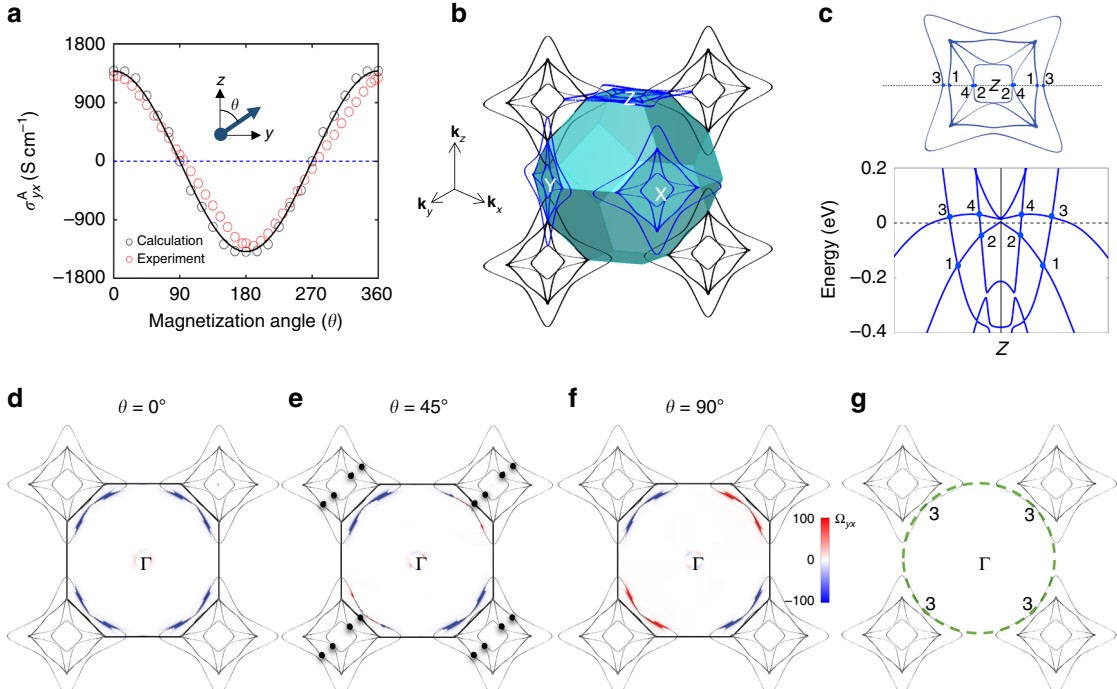

**Fig. 3 The anomalous Hall effect and topological band structure. a** The anomalous Hall conductivity $\sigma_{yx}^A$ as rotating the magnetization from the $z$-axis to $y$-axis. The experimental and theoretical results are represented by red and black circles, respectively. The solid curve is a cosine profile to guide eyes. **b** Nodal rings and the first Brillouin zone of $Co_2MnAl$. Without SOC, there are nodal rings in these mirror planes, $k_{x,y,z} = 0$ or $\pi$. For example, there are four nodal rings that are labeled as #1–4 in **c**, centered at the $Z$ point of the FCC Brillouin zone. When magnetization align along the [001] direction by SOC, all nodal rings are gapped out because of the mirror symmetry breaking, except those in the $k_z = 0$ or $\pi$ plane. The preserved nodal rings and related band structure are shown in **c**. The band dispersion, in which the nodal ring crossing points are indicated by blue dots, is along the black dotted line in the upper panel. **d–f** The Berry-curvature ($\Omega_{yx}$) distribution in the $k_x = 0$ plane centered at $\Gamma$ for different magnetization angle ($\theta$). Nodal rings (black) are gapped out in this plane. Near the charge neutral point, the Berry curvature is mainly contributed by the gapped nodal ring #3. When $\theta = 45°$, nodal rings #1 and #4 evolve into Weyl points (filled black circles in **e**), while nodal rings #2 and #3 are fully gapped. **g** The nodal ring #3 at four corners can be treated as the reconstruction of a large ring, which is illustrated by a green dashed circle, centered at $\Gamma$.

topological bands, the large AHE is predominantly contributed by the topological carriers.

We can control the AHE response by rotating the magnetization axis, when fixing the current and Hall voltage probes on the (001) plane (see the right inset of Fig. 2a). In the experiment, we observed a $\cos(\theta)$-like oscillation of $\sigma_{yx}^A$ with respect to the magnetization angle, $\theta$. As shown in Fig. 3a, the experiment is consistent with our theoretical calculations. As the magnetization changes from [001] to [010], nodal rings appear in the the $k_y = 0$ plane, while they disappear in $k_{x,z} = 0$ planes. In between [001] and [010], #3 nodal rings are fully gapped in all three planes, although some other nodal rings (e.g., #1 and #4) evolve into Weyl points (Fig. 3e). As a consequence, the induced Berry curvature $\Omega_{yx}$ evolves dramatically from [001] to [010], as shown in Fig. 3d, f, leading to the oscillating $\sigma_{yx}^A$ as seen in experiments. The nearly cosine-like profile is consistent with the fact that $Co_2MnAl$ exhibits weak anisotropy in the magnetization orientation in experiment. We should point out that the present $\cos(\theta)$-type AHE is different from the known planar Hall effect, which has a $\sin(2\theta)$ dependence[57]. The planar Hall effect appears in an FM metal where the magnetization remains unchanged as rotating an in-plane magnetic field, or a nonmagnetic WSM where the chiral anomaly plays a role[58,59]. The gap opening of the various nodal lines and the associated band structure reconstruction driven by the rotation of magnetization is manifested by the angular dependence of Hall coefficient $R_H$. In Supplementary Fig. 7, we present the $\rho_{xy}$ data collected under various magnetization orientations between [001] and [010]; $R_H$ can be extracted from the linear fit in the field range where magnetization is saturated for

each orientation. We found that $R_H$ varies remarkably as the magnetization is rotated from [001] to [011], with a sign change near 20°, as shown in the right inset of Fig. 2a. Moreover, $R_H$ for $M \parallel$ [111] is almost four times of that for $M \parallel$ [001], see the left inset of Fig. 2a. These observations, together with the angular dependence of $\sigma_{yx}^A$ shown in Fig. 3a, is consistent with the theoretical picture of the band structure and topology tuning by the rotation of magnetization in $Co_2MnAl$ discussed above.

In principle, a simple nodal ring can evolve into a pair of Weyl points when the magnetization breaks the corresponding mirror symmetry, as illustrated in Fig. 4a. The induced AHC is proportional to the Weyl point separation[12,13]. This scenario is seemingly consistent with our observation on nodal rings #1 and #4. However, we do not observe the existence of Weyl points for nodal ring #3, which is the main source of the AHE observed in the experiment, when rotating the magnetization axis.

Actually the missing of Weyl points for nodal ring #3 can explain the large magnitude of the AHC observed in $Co_2MnAl$. Take the [001] magnetization as an example. The theoretical value of $\sigma_{yx}^A \sim 1400 \, \Omega^{-1} \, cm^{-1}$ is as large as the AHC of a 3D QAHE[60], $\frac{2e^2}{h} \frac{1}{a} = 1347 \, \Omega^{-1} \, cm^{-1}$, where $a$ takes the experimental value 5.75 Å. We note that this is not coincidence. Although this nodal ring is centered at the $Z$ point, it emerges from a large nodal ring that is centered at $\Gamma$. Because it even crosses the Brillouin zone boundary, the $\Gamma$-centered ring gets reconstructed to smaller rings at the zone boundary (see Fig. 3g). Although smaller nodal rings generate Weyl points, a reconstructed nodal ring does not generate Weyl points (illustrated in Fig. 4c). This is because a Weyl point, if it exists, must annihilate with another one from the

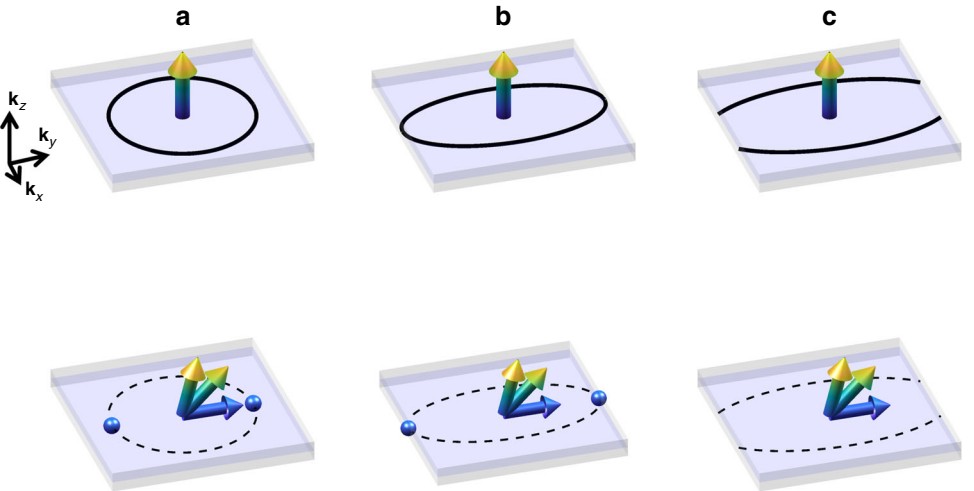

**Fig. 4 Schematic transition between nodal rings and Weyl points. a** The nodal ring (solid ring) is protected by the in-plane mirror symmetry where the spin (arrow) points out of plane, in the upper panel. If the spin rotates to break the mirror symmetry, the nodal ring gets gapped out (dashed ring), giving rise to a pair of Weyl points (blue spheres). The induced AHE is proportional to the separation of the Weyl points, that is, the diameter of the nodal ring ($k_d$) in a form $\frac{e^2}{h}\frac{k_d}{2\pi a}$, where $a$ is the lattice parameter. **b** The nodal ring is as large as the Brillouin zone size, as a critical point. The resultant Weyl point is pushed to the zone boundary and meets another Weyl point with opposite chirality from the second zone. The induced AHE is quantized to $\frac{e^2}{h}\frac{1}{a}$. **c** The nodal ring is larger than the Brillouin zone size, as an open nodal ring. Then, the gapped ring does not induce Weyl points in the open direction, generating a 3D quantized Hall conductivity, $\frac{e^2}{h}\frac{1}{a}$.

neighboring Brillouin zone. As a consequence, the band structure has a gap and the induced AHC has a quantized value $\frac{e^2}{h}\frac{1}{a}$. In our material, there are two groups of gapped nodal rings on the $\mathbf{k}_{x,y} = 0$ planes. Therefore, the ideal AHC is $\frac{2e^2}{h}\frac{1}{a}$. Because the nodal ring cannot be fully gapped out in the Fermi energy and there are also contribution for other bands, we do not expect a true 3D QAHE here. We note that the same scenario can also be applied to $Co_2MnGa$. In addition, if the $\Gamma$-centered ring was smaller than the Brillouin zone size in $Co_2MnAl$, it could generate Weyl points but smaller AHC than the quantized value. This is actually the case of Heusler compounds $Fe_2MnX$ ($X =$ P, As, Sb)[46], $Co_2VGa$[45], and $Co_2TiX$ ($X =$ Si, Ge, Sn)[43]. We can rationalize these full Heusler magnets in the same framework here.

## Conclusions

Based on the understanding of $Co_2MnAl$, we can obtain useful insights to design magnetic materials with strong AHE. To generate large AHC, we need multiple large nodal rings that are larger than the Brillouin zone size. To host multiple nodal rings (in the absence of SOC), we prefer to have multiple mirror planes in the crystal structure, which usually indicates high-index space groups. In general, highly symmetric magnetic materials with strong SOC are an optimal choice, not only for the AHE, but also for other Berry-curvature-induced phenomena like the anomalous thermal Hall effect and anomalous Nernst effect. Similarly, nonmagnetic materials with many mirror planes and strong SOC are ideal for the spin Hall effect, as pointed out in ref. [61].

Our theory and experiment together also demonstrate that the form of the gapless nodes (Weyl points or nodal rings), as well as their locations and velocities, all strongly rely on the magnetization axis. Therefore, the magnetization can potentially be utilized to engineer an artificial gauge field[62,63] and spacetime geometric structure[64,65] of Weyl fermions. On the other hand, Weyl fermions can also mediate a strong coupling between magnetization dynamics and electromagnetic fields[63], thus allowing for the electric control of magnetic dynamics in this system. Thus, $Co_2MnAl$ provides an ideal intrinsically magnetic platform to explore the interplay between magnetic dynamics and topological physics for the development of a new generation of spintronic devices.

## Methods

**Experimental.** $Co_2MnAl$ single crystals were grown using FZ technique. The material rods used for the FZ growth were made using an induction furnace. The crystals were annealed at 1550 K for 1 week, followed by annealing at 873 K for another week. The powder X-ray diffraction measurements confirmed that the $Co_2MnAl$ single crystals grown using the above method possesses a cubic structure. Since $Co_2MnAl$ has multiple structural phases with different types of disorders as mentioned in the main text and only the $L2_1$ phase is predicted to host an FM WSM phase, we performed detailed TEM analyses to identify its structure phase. The TEM sample was prepared using a Thermo Fisher Helios NanoLab Dual-Beam focused ion beam system. The HAADF-STEM images are taken with the Thermo Fisher Titan S/TEM equipped with a spherical aberration corrector, and it was operating at 300 kV accelerating voltage with a probe convergence angle of 30 mrad. Magnetization properties were measured using a SQUID (VSM, Quantum Design). The electrical transport properties were measured using a standard four probe method in a physical property measurement system (Quantum Design).

**Ab initio calculation.** We have performed density-functional theory (DFT) calculations with a full-potential local-orbital minimum-basis (FPLO)[66] code to calculate electronic structure. The exchange and correlation energies are considered in the generalized gradient approximation by Perdew–Burke–Enzerhof scheme[67]. SOC is included. From the DFT electronic structure, we have projected the Bloch wave function into Wannier functions (Co-3$d$, Mn-3$d$, and Al-3$p$ orbitals), to construct an effective Hamiltonian ($\hat{H}$) for the bulk material.

We have evaluated the AHC ($\sigma_{yx}^A$) and Berry curvature ($\Omega_{yx}$) by the Kubo-formula approach in the linear response scheme[9],

$$\sigma_{ij}^A(\mu) = -\frac{e^2}{\hbar}\int d\xi \frac{\partial f(\xi - \mu)}{-\partial \xi}\int_{BZ}\frac{d\mathbf{k}}{(2\pi)^3}\sum_{\epsilon_n < \mu}\Omega_{ij}^n(\mathbf{k}), \quad (1)$$

$$\Omega_{ij}^n(\mathbf{k}) = i\sum_{m \neq n}\frac{\langle n|\hat{v}_i|m\rangle\langle n|\hat{v}_j|m\rangle - (i \leftrightarrow j)}{(\epsilon_n(\mathbf{k}) - \epsilon_m(\mathbf{k}))^2}. \quad (2)$$

Here $\epsilon_n$ is the eigenvalue of the $|n\rangle$ eigenstate, and $\hat{v}_i = \frac{d\hat{H}}{\hbar dk_i}$ ($i = x, y, z$) is the velocity operator, $\mu$ is Fermi level of the system, and $f(\xi - \mu)$ is the Fermi–Dirac distribution. A $k$-point of grid of $300 \times 300 \times 300$ is used for the numerical integration in Eq. (1).

## Data availability

All relevant data are available from the corresponding authors upon request.

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

## Acknowledgements

The experimental work at Tulane and Penn State is supported by the US National Science Foundation under Grant DMR1917579. L.Miao, L.Min, and N.A.'s work is supported by the Penn State Center for Nanoscale Science, an NSF MRSEC under the Grant number DMR-1420620. B.Y. acknowledges the financial support by the Willner Family Leadership Institute for the Weizmann Institute of Science, the Benoziyo Endowment Fund for the Advancement of Science, Ruth and Herman Albert Scholars Program for New Scientists, and the European Research Council (ERC Grant no. 815869). C.-X.L. acknowledges the support of the Office of Naval Research (Grant No. N00014-18-1-2793), the US Department of Energy (Grant no. DESC0019064), and Kaufman New Initiative research grant KA2018-98553 of the Pittsburgh Foundation. Y.Z. and Y.W. acknowledge financial support from the National Science Foundation through the Penn State 2D Crystal Consortium-Materials Innovation Platform (2DCC-MIP) under NSF cooperative agreement DMR-1539916.

## Author contributions

Z.M. and B.Y. conceived the project. P.L., Y.W., and Y.Z. synthesized the single crystals. J.L. made the feed rods for single crystal growth. P.L. and W.N. performed the transport and magnetization measurements. L.Min oriented crystals using Laue pattern. L.Miao and N.A. carried out TEM analyses. J.K. performed theoretical calculations. C.-X.L. discussed the gauge field and magnetic dynamics. P.L., J.K., Z.M., and B.Y. wrote the manuscript. All authors examined results. The experimental and theoretical parts of this work are supervised by Z.M. and B.Y., respectively.

## Competing Interests

The authors declare no competing interests.
