## [Peer Review File · Nature Communications]

Reviewers' comments:

Reviewer #1 (Remarks to the Author):

Peigang Li et al. report the tunable and large anomalous Hall effect (AHE) on a ferromagnetic topological semimetal, Co₂MnAl, in which the band topology of the spin-polarized bands plays an important role. Combining measurements on magnetotransport properties of a single crystal and band structure calculations, the authors claimed that the observed large AHE is induced by the annihilation of Weyl points in a certain magnetic field direction and the resulting large Berry curvature. Magnetic Heusler compounds have been intensively studied both theoretically and experimentally these days as a promising material class showing topological properties. In this regard, the current work is timely and may provide important clues for realizing spintronics based on topological materials.

As described above, I found that the main claims in this work are (1) observation of large anomalous Hall effect and (2) its tunability on a rotating magnetic field. For the large anomalous Hall effect in Co₂MnAl, it has already been recognized in 2012 on the basis of Berry curvature calculations [Kuebler et al Phys. Rev. B 85, 012405 (2012)] and was found to be in good agreement with the experiments [Vilanova Vidal et al, Appl. Phys. Lett. 99 132509 (2011)]. Recently, this aspect has also been emphasized in review papers on Heusler compounds, for example, K. Manna et al. Nat. Rev. Mater. 3 244 (2018). While the authors used a good-quality single crystal of Co₂MnAl, rather than the thin film, the claim is essentially the same.

In topological magnets, the Berry curvature and thus the AHE are well known to be sensitive to the orientation of magnetization, as found in the previous theoretical studies [J. Kuebler et al. Europhys. Lett. 114, 47005 (2016), Z. Wang et al. Phys. Rev. Lett. 117, 236401 (2016), S. Chadov et al. Phys. Rev. B 96, (2017)]. Therefore, the novelty of this work lies in the experimental observations of the tunable AHE. The essential question is then whether or not the experimental results indeed confirm the theoretical prediction.

In Fig. 3(a), the authors present good agreement between experiments and calculations as evidence. However, I have a strong doubt on this comparison. The experimental and theoretical results show that the anomalous Hall conductivity, σ_{yx} , follows the $\cos(\theta)$ dependence as a function of magnetization orientation (θ). This angle dependence can be simply understood by the variation of the normal component of magnetization M_z , without considering the change in the Berry curvature. In fact, such a $\cos(\theta)$ dependence of AHE is well known for many ferromagnetic metals. Therefore, it is not at all clear to me which part of experimental or theoretical data is the evidence of the Berry curvature change due to the annihilation of Weyl points. As mentioned above, I think this question is the most critical one and should be clarified.

In summary, I found that the proposed mechanism of tunable AHE is not confirmed by the experimental results, presented in this work. I do not recommend publication of this work in Nature Communications.

Reviewer #2 (Remarks to the Author):

In their manuscript, Li et al. study the anomalous Hall effect in the candidate ferromagnetic Weyl semimetal material Co₂MnAl under magnetic field. The paper contains experimental data on the crystal structure as well as temperature and field dependent magnetization and electrical transport. It is supplemented with an extensive theoretical analysis based on topological aspects of the band

structure. The major finding is a giant, room temperature Hall response at finite field, which is attributed to the field induced reconfiguration of electronic bands.

This finding is clearly interesting. Moreover, data and theory appear sound and plotting them on top of each others matches remarkably well (Fig. 3a). However, at present I can not recommend the paper for publication, as it fails to provide strong evidence for its conclusions (this is markedly one of the main criteria for publication in Nature Comm.). In particular, the presented transport data by no means supports the intricate theoretical picture which is elaborated on in the second part of the manuscript.

Detailed comments:

(1) If the proposed theoretical scenario indeed holds, the partial opening of gaps (energy scales seem to be 200 - 2000 K, according to Fig S3) at the various nodal lines and the associated reconfiguration of band structure and Fermi surfaces should reflect itself in various other experimental observables: quantum oscillations, thermodynamics, photoemission to name a few examples. A detailed analysis of at least one of those probes is unavoidable to "demonstrate a way to modify the band topology via a weak magnetic field", as advertised in the abstract.

(2) The authors highlight that their material is a "ferromagnet", even in the title. A ferromagnet should have a remnant magnetization at zero field, even soft ferromagnets do. I strongly encourage the authors to include a hysteresis curve or to rephrase their statements (discuss domains etc.) to avoid further confusion.

(3) For purists, the analogue also holds for the word "anomalous Hall effect": It describes the remnant Hall response at zero field. I acknowledge that the vague attribute "anomalous" seems to leave some freedom of interpretation, but explanations are necessary. A theoretical working definition could be that the field-caused-Berry-curvature induces a Hall response which by far exceeds the orbital effect at the same field strength — is this the case?

(4) It is incorrect to conclude from " σ_{yx} hardly varies with σ_{xx} " that the anomalous Hall effect is intrinsic. In fact, it is well known that under rather generic circumstances the scattering rate drops out even in extrinsic contributions (side jump, and skew scattering in models of Gaussian disorder) to the Hall response. In order to estimate whether the effect is intrinsic or extrinsic, I would rather resort to the temperature dependence of the data: For $I \parallel (111)$, it seems that Hall response doesn't care whether scattering is phonon- or impurity dominated. (A plot of $\rho_{xx}(T)$ in the supplement would help the reader to get a feeling at which temperature the crossover occurs.) For $I \parallel (001)$, the situation seems less obvious and the negative slope of σ_{yx} needs explanation.

(5) The common definition of the Hall angle is the arcus tangent of the Hall angle which the authors define. At small angles it doesn't matter, but since the angle is so high, it probably does.

(6) Multiple typos could be corrected now, or in the proofs: "are are", "gaping", "gaped", "this is because that".

Reviewer #3 (Remarks to the Author):

In the manuscript "Giant room temperature anomalous Hall effect and magnetically tuned topology in the ferromagnetic Weyl semimetal Co₂MnAl", P. Li et al. have presented large anomalous Hall effect and Hall angle up to room temperature, which they attribute to the Berry curvature induced by non-trivial band topology. Then they propose a mechanism to generate large anomalous Hall effect by gapping nodal rings without requiring the existence of Weyl points.

Single crystal growth has helped reveal one of the highest values of anomalous Hall effect and the Hall

angle at room temperature. The theoretical calculations support the observation of the Hall effect, and this is a topical subject. I think the results presented here will be of interest to the readers in the field. However, there are few things that I would like to see explained.

1) I think separation of anomalous Hall effect due to simple ferromagnetic contribution (say if the band structure was that of a simple metal) would be helpful to better understand the actual contribution coming from the non-trivial band structure. It may also help to see if the anomalous Hall conductivity value is intrinsically related to the Berry curvature or has some contribution from the ferromagnetic ordering. It may be that the ferromagnetic Hall conductivity is small compared to the one observed here. But separating these two contributions will help readers. In other words, separating normal, anomalous (proportional to the magnetization) and topological Hall components would make it more clear.

2) I would like to see the discussion about how different the proposed mechanism for the large Hall effect presented here is from that discussed in GdPtBi by Joe Checkelsky's group [Reference 1 in supplementary materials)?

3) Experimental data presented in Fig. 3(a) needs more details. How is the magnetization angle changed experimentally? If it is by tilting the magnetic field, then what is the relation between current, magnetic field and voltage?

4) It would be helpful to see ρ_{xx} vs H and ρ_{xx} vs T . What is the RRR value of the measured crystal etc.

5) I prefer candidate Weyl (semi)metal in the title as opposed to Weyl semimetal at this point unless supported by other experimental results for the Weyl (semi)metallic behavior. I think showing band structure over the whole Brillouin zone would be helpful (either in the main text or even in supplementary section. Without seeing a full band structure I am not sure whether to call it a metal or a semimetal. Are there any other experimental evidence to call it a semimetal?

Response to Reviewer 1

We thank the reviewer for taking time to review our manuscript. The reviewer's comments have been very helpful in improving our manuscript. We have now addressed the issues raised in the Reviewer's report (marked in black below) and made point-to-point response (in blue) to the criticism.

Peigang Li et al. report the tunable and large anomalous Hall effect (AHE) on a ferromagnetic topological semimetal, Co_2MnAl , in which the band topology of the spin-polarized bands plays an important role. Combining measurements on magnetotransport properties of a single crystal and band structure calculations, the authors claimed that the observed large AHE is induced by the annihilation of Weyl points in a certain magnetic field direction and the resulting large Berry curvature. Magnetic Heusler compounds have been intensively studied both theoretically and experimentally these days as a promising material class showing topological properties. In this regard, the current work is timely and may provide important clues for realizing spintronics based on topological materials.

Response: We appreciate the referee's positive assessment of our work.

As described above, I found that the main claims in this work are (1) observation of large anomalous Hall effect and (2) its tunability on a rotating magnetic field. For the large anomalous Hall effect in Co_2MnAl , it has already been recognized in 2012 on the basis of Berry curvature calculations [Kuebler et al Phys. Rev. B 85, 012405 (2012)] and was found to be in good agreement with the experiments [Vilanova Vidal et al, Appl. Phys. Lett. 99 132509 (2011)]. Recently, this aspect has also been emphasized in review papers on Heusler compounds, for example, K. Manna et al. Nat. Rev. Mater. 3 244 (2018). While the authors used a good-quality single crystal of Co_2MnAl , rather than the thin film, the claim is essentially the same.

Response:

(1) Theory aspect. The main claims, as we stressed in the abstract, are (1) the tunability of the topology, i.e. the Weyl point - Nodal line transformation, and (2) the strong AHE induced mainly by the nodal line gapping instead of the Weyl points. Kuebler and Felser predicted the large AHE for Co_2MnAl in 2012 and then pointed out the existence of Weyl points in 2016 [EPL 114, 47005 (2016)], which was reviewed recently [K. Manna et al. Nat. Rev. Mater. 3 244 (2018)]. As far as we understood, these earlier works have not revealed the evolution between Weyl points and nodal lines and the tunability, and also attributed the AHE to Weyl points. In the revised manuscript, we have added the citation of the paper-2012 and the recent review-2018 in the references.

(2) Regarding our experimental work, one important issue was not made clear in our original manuscript, i.e. the APL-2011 thin film (B2 phase) is structurally different from the single crystal ($L2_1$ phase) reported in our work. This probably misled the referee's judgment. As indicated in the manuscript, Co_2MnAl can have several structural phases with different disorder types, including $L2_1$ (Mn, Al ordered), B2 (Mn, Al disordered), DO_3 (Co, Mn disordered), A2 (Co, Mn, Al disordered). Among these structural phases, only the $L2_1$ phase is predicted to host a ferromagnetic Weyl state. Our Co_2MnAl single crystals have been confirmed to possess the $L2_1$ structural phase. However, the Co_2MnAl polycrystalline thin films synthesized by Vidal et al were

found to have the B2 phase and show DO3 disorder after annealing. Therefore, the anomalous Hall effect probed on such films cannot be used for comparison with the intrinsic anomalous Hall effect predicted by theory for the L2₁ phase. In this sense, our observation of large intrinsic anomalous Hall conductivity and large anomalous Hall angle on Co₂MnAl single crystals with the L2₁ phase provides the first experimental evidence for the prediction of large Berry curvature of Co₂MnAl. The other importance of our work lies in that it advances understanding of band topology tuning by magnetization as discussed above. In the revised manuscript, we clarified the structural difference between our Co₂MnAl single crystals and the Co₂MnAl thin films reported by Vidal et al.

In topological magnets, the Berry curvature and thus the AHE are well known to be sensitive to the orientation of magnetization, as found in the previous theoretical studies [J. Kuebler et al. Europhys. Lett. 114, 47005 (2016), Z. Wang et al. Phys. Rev. Lett. 117, 236401 (2016), S. Chadov et al. Phys. Rev. B 96, (2017)]. Therefore, the novelty of this work lies in the experimental observations of the tunable AHE. The essential question is then whether or not the experimental results indeed confirm the theoretical prediction.

In Fig. 3(a), the authors present good agreement between experiments and calculations as evidence. However, I have a strong doubt on this comparison. The experimental and theoretical results show that the anomalous Hall conductivity, σ_{yx} , follows the $\cos(\theta)$ dependence as a function of magnetization orientation (θ). This angle dependence can be simply understood by the variation of the normal component of magnetization M_z , without considering the change in the Berry curvature. In fact, such a $\cos(\theta)$ dependence of AHE is well known for many ferromagnetic metals. Therefore, it is not at all clear to me which part of experimental or theoretical data is the evidence of the Berry curvature change due to the annihilation of Weyl points. As mentioned above, I think this question is the most critical one and should be clarified.

Response:

(1) Theory aspect. We have further clarified the novelty of our theory and its distinction from the previous work, in the response to the first comment. We should point out that the believed $\cos(\theta)$ -profile of the AHE is an approximated scenario. It assumes that the material is ideally isotropic, although this assumption is not always solid in reality. As demonstrated in our Fig. 3a, our theoretical and experimental results deviate slightly from the ideal $\cos(\theta)$. This is related to the anisotropy even in a cubic material. Regarding the comment “*which part of experimental or theoretical data is the evidence of the Berry curvature change*”, we actually provide clear theoretical evidence for the “*the Berry curvature change*” as rotating the magnetic field (see Fig. 3d-3f). The quantitative agreement between theory and AHE experiment supports the Berry curvature change. The AHE is predicted to be large by summing over the total Berry curvature. Beyond the brutal-force computation of the Berry curvature, our work revealed a deep, but simple mechanism beyond the large magnitude AHE. It originates from the fully gapped nodal lines and thus can reach as large as the quantized Hall conductance, $2e^2/h/a$. This mechanism leads to practical recipes to find AHE materials.

(2) Experiment aspect I. The magnetic field is commonly believed to affect the band structure as a perturbation. This is also part of the assumption of the $\cos(\theta)$ dependence. However, we provide further experimental evidences for the carrier density variation induced by band structure change in the following. Since our theory predicts the band structure of Co_2MnAl depends on the orientation of magnetization, the carrier density is expected to vary as the magnetization is rotated. For instance, if the magnetization is along the [001] direction, four nodal rings on the $k_z=0$ planes are not gapped due to the preserved mirror symmetry on this plane, while the nodal rings at the $k_x=0$ and $k_y=0$ planes are gapped out. In contrast, when the magnetization is rotated to [111], all the nodal rings on $k_x=0$, $k_y=0$ and $k_z=0$ planes are gapped out. Therefore, we expect to observe a different carrier density when the magnetization orientation is changed from [001] to [111]. This is indeed manifested by our experiments. As shown in Fig. 1a (for referee) attached below, the Hall coefficient (the Lorentz-force part) probed for $B//[111]$ is almost four times of that for $B//[001]$, suggesting the carrier density for $B//[111]$ is just one quarter of that for $B//[001]$. Moreover, we find that R_H varies remarkably as the magnetization is rotated from [001] to [011], with a sign change near 20° , as shown in Fig. 1b below. These observations clearly indicate the band structure of Co_2MnAl varies with the change of magnetization orientation. This provides additional support for our theoretical prediction that the Berry curvature changes with the rotation of magnetization. We have added the Hall coefficient data shown in Fig. 1 (for referee) and relevant discussions to the revised manuscript.

Figure 1 (for referee). (a) Hall resistivity as a function of magnetic field at 300K for the two samples shown in Fig. 2 in the manuscript, probed with $B//[001]$ and $B//[111]$ respectively. The data collected with $B//[001]$ are offset for clarity. (b) Hall coefficient as a function of the orientation angle of magnetization for the sample with $I//[001]$. The raw data of $\rho_{xy}(B, \theta)$ from which the data in this panel are extracted are presented in the supplementary materials (SM). The inset shows the experimental set-up.

(3) Experiment aspect II. As we showed in Fig. 2b, the intrinsic anomalous Hall conductivity (AHC) for $I//[001]$ has a stronger temperature dependence than that of $I//[111]$. This is also a significant consequence of the band structure change. As shown in supplementary Fig. S1, the Berry curvature exhibits different chemical potential dependence between (001) and (111), caused by the band structure change. As a consequence, AHC of (001) shows a negative slope as increasing temperature while that of (111) is relatively temperature insensitive, which is related by the temperature broadening of the Berry curvature by the Fermi-Dirac distribution. (See also the reply to Reviewer 2). We have made this point clear in the revised manuscript.

In summary, I found that the proposed mechanism of tunable AHE is not confirmed by the experimental results, presented in this work. I do not recommend publication of this work in Nature Communications.

Response: We hope we have satisfactorily addressed the questions the referee raised above.

Response to Reviewer 2

In their manuscript, Li et al. study the anomalous Hall effect in the candidate ferromagnetic Weyl semimetal material Co_2MnAl under magnetic field. The paper contains experimental data on the crystal structure as well as temperature and field dependent magnetization and electrical transport. It is supplemented with an extensive theoretical analysis based on topological aspects of the band structure. The major finding is a giant, room temperature Hall response at finite field, which is attributed to the field induced reconfiguration of electronic bands.

This finding is clearly interesting. Moreover, data and theory appear sound and plotting them on top of each other matches remarkably well (Fig. 3a). However, at present I cannot recommend the paper for publication, as it fails to provide strong evidence for its conclusions (this is markedly one of the main criteria for publication in Nature Comm.). In particular, the presented transport data by no means supports the intricate theoretical picture which is elaborated on in the second part of the manuscript.

Response: We thank the referee for taking time to review our manuscript and giving positive assessment of our work. We also appreciate the insightful comments made by the referee, which have been very helpful in improving our manuscript. We give point-to-point response to each of the referee's comments below.

Detailed comments:

(1) If the proposed theoretical scenario indeed holds, the partial opening of gaps (energy scales seem to be 200 - 2000 K, according to Fig. S3) at the various nodal lines and the associated reconfiguration of band structure and Fermi surfaces should reflect itself in various other experimental observables: quantum oscillations, thermodynamics, photoemission to name a few examples. A detailed analysis of at least one of those probes is unavoidable to “demonstrate a way to modify the band topology via a weak magnetic field”, as advertised in the abstract.

Response: We agree with the referee that the partial gap opening of the various nodal lines and the associated band structure reconstruction should be manifested by other experimental variables like quantum oscillations, thermodynamics and photoemission. However, these experimental probes are not feasible for Co_2MnAl . We have capability of doing specific heat measurements. Nevertheless, since Co_2MnAl is ferromagnetic with its Curie temperature above 700K, specific heat measurements under magnetic fields are impossible since the magnetization of the sample would cause magnetic torque under magnetic fields, which can break the sample stage. We have also considered quantum oscillation measurements. But the observation of quantum oscillations

requires high mobility; this does not seem satisfied in our samples since its transport carriers are dominated by normal carriers instead of the Weyl fermions as discussed in the manuscript. We did not observe quantum oscillations in magnetoresistance up to 9T [see Fig. 2c (for referee) attached below]. Although ARPES is a direct probe for band structure, ARPES measurements under magnetic fields have not been available in any labs and this is also beyond our expertise.

Given these challenges, we have attempted extracting the information on the band structure reconstruction through analyzing the variation of Hall coefficient with the magnetization orientation, to monitor the carrier density change. Our theory presented in the manuscript shows the gap opening of nodal lines depend on the magnetization orientation. For instance, when the magnetization is along the [001] direction, four nodal rings on the $k_z=0$ planes are not gapped due to the preserved mirror symmetry on this plane, while the nodal rings at the $k_x=0$ and $k_y=0$ planes are gapped out. In contrast, when the magnetization is along the [111] direction, all the nodal rings on $k_x=0$, $k_y=0$ and $k_z=0$ planes are gapped out due to broken mirror symmetries on all these planes. Given such a band structure difference between $M//[001]$ and $M//[111]$, different carrier density can naturally be expected between these two magnetization orientations. This is indeed probed in our experiments. The carrier density can generally be evaluated from Hall coefficient R_H . For ferromagnetic materials, Hall resistivity can be expressed as $\rho_{xy} = R_H B + 4\pi R_s M$ and R_H can be estimated from the linear slope of ρ_{xy} in the field range where M is saturated. As shown in Fig. 2a attached below, R_H for $M//[111]$ is almost four times of that for $M//[001]$, indicating that carrier density for $M//[001]$ is four times higher than that for $M//[111]$. Moreover, we find that R_H varies remarkably as the magnetization is rotated from [001] to [011], with a sign change near 20° , as shown in Fig. 2b below. These observations, together with the dependence of anomalous Hall conductivity and anomalous Hall angle on the magnetization orientation shown in Fig. 2a & 2b in the manuscript, provide strong support for our argument that the band structure and topology of Co_2MnAl can be tuned by the rotation of magnetization. We have added these new data and relevant discussions to the revised manuscript and the raw data of $\rho_{xy}(B, \theta)$ measured on the $I//(001)$ sample to the supplementary materials.

Figure 2 (for referee 2). (a) Hall resistivity as a function of magnetic field at 300K for the two samples shown in Fig. 2 in the manuscript, probed with $B//[001]$ and $B//[111]$ respectively. The data collected with $B//[001]$ are offset for clarity. (b) Hall coefficient as a function of the orientation angle of magnetization for the sample with $I//(001)$. The raw data of $\rho_{xy}(B, \theta)$ from which the data in this panel are extracted are

presented in SM. The inset shows the experimental set-up. (c) Longitudinal magnetoresistance, $MR = [\rho_{xx}(H) - \rho_{xx}(0)] / \rho_{xx}(0)$ as a functional of magnetic field for Co_2MnAl .

Furthermore, the band structure change tuned by magnetization orientation is also manifested by different temperature dependence of the anomalous Hall conductivity between $M//[001]$ and $M//[111]$ cases (Fig. 2b in the manuscript). See more discussions in the response to Question (4).

(2) The authors highlight that their material is a “ferromagnet”, even in the title. A ferromagnet should have a remnant magnetization at zero field, even soft ferromagnets do. I strongly encourage the authors to include a hysteresis curve or to rephrase their statements (discuss domains etc.) to avoid further confusion.

Response: We thank the referee for this suggestion. Our Co_2MnAl single crystal samples does show hysteresis though it is a soft ferromagnet. The remnant magnetization is 0.1 and 0.2 $\mu_B/\text{f.u.}$ at 2K, respectively, for $B_{\perp}(001)$ and $B_{\perp}(111)$. We have added these data into Figure 1e as insets in the revised manuscript, as shown below.

Figure 3 (for referee 2). Isothermal magnetization at 2 K and 400 K under two different field orientations for Co_2MnAl . The insets show the enlargement of the magnetization around zero fields.

(3) For purists, the analogue also holds for the word “anomalous Hall effect”: It describes the remnant Hall response at zero field. I acknowledge that the vague attribute “anomalous” seems to leave some freedom of interpretation, but explanations are necessary. A theoretical working definition could be that the field-caused-Berry-curvature induces a Hall response which by far exceeds the orbital effect at the same field strength — is this the case?

Response: Yes, anomalous Hall effect (AHE) refers to the zero-field Hall response, as we commonly extract the AHE resistivity from $\rho_{xy} = R_{HB} + \rho_{Axy}$. We have added the clarifications in the revised manuscript.

We would rephrase “*the field-caused-Berry-curvature*” to “*that the field-tuned-Berry-curvature*”. Here the magnetic field rotates the magnetization. Then, the magnet moments’ rotation changes the Berry curvature by coupling the exchange field to the band structure. The direct coupling (Zeeman split) between the field and band structure is negligible here. One can find that only the field direction, rather than the magnitude, affects the AHE. We added related discussions to the revised manuscript.

(4) It is incorrect to conclude from “ σ_{yx} hardly varies with σ_{xx} ” that the anomalous Hall effect is intrinsic. In fact, it is well known that under rather generic circumstances the scattering rate drops out even in extrinsic contributions (side jump, and skew scattering in models of Gaussian disorder) to the Hall response. In order to estimate whether the effect is intrinsic or extrinsic, I would rather resort to the temperature dependence of the data: For $I \parallel (111)$, it seems that Hall response doesn’t care whether scattering is phonon- or impurity dominated. (A plot of $\rho_{xx}(T)$ in the supplement would help the reader to get a feeling at which temperature the crossover occurs.) For $I \parallel (001)$, the situation seems less obvious and the negative slope of σ_{yx} needs explanation.

Response: We thank the referee for this suggestion and have removed the inset (σ_{yx} vs σ_{xx}) of Fig. 2b in the revised manuscript. Following the referee’s suggestion, we have also added the $\rho_{xx}(T)$ data for both $I \parallel (001)$ and $I \parallel (111)$ to the SM and also attach it below. $\rho_{xx}(T)$ exhibits clear temperature dependence and localization behavior for both (001) and (111), indicating the scattering rate varies with temperature. Compared to ρ_{xx} , corresponding σ_{yx} remains nearly unchanged from several K to 300 K. As the referee pointed out, this is a proof of the intrinsic nature of the AHE.

For $I \parallel (001)$, σ_{yx} exhibits a slightly negative slope as compared to the $I \parallel (111)$ case. This can be rationalized as temperature-dependence of the Berry curvature due to the Fermi-Dirac distribution (see Eq. S1). As we showed in Fig. S1a, σ_{yx} of $I \parallel (001)$ has a different chemical potential μ dependence from that of $I \parallel (111)$. This is a direct consequence of the band structure change. Below $\mu = 0$, σ_{yx} of (001) is obviously smaller than σ_{yx} of (111). Then the (001) case drops quicker than the (111) case upon increasing temperature (Fig. S1b), since temperature-broadening averages the Berry curvature in a window around $\mu = 0$. This is consistent with our experiment.

Figure 4 (for referee 2). Temperature dependence of resistivity for two samples with I// (001) and I//(111).

(5) The common definition of the Hall angle is the arcus tangent of the Hall angle which the authors define. At small angles it doesn't matter, but since the angle is so high, it probably does.

Response: We agree with the referee that the exact definition of Hall angle θ_H should be

$\tan^{-1}(\sigma_{yx}/\sigma_{xx})$ and defining σ_{yx}/σ_{xx} as a Hall angle is inappropriate for large angles. Nevertheless, we note that most previously reported works for magnetic topological semimetals use the definition of σ_{yx}/σ_{xx} despite large angles (e.g. see Liu et al., Nature Physics 14, 1125 (2018)). In order to compare with those previous results, we followed the same definition. In the revised manuscript, we have adopted the definition of $\tan\theta_H = \sigma_{yx}/\sigma_{xx}$ when we give discussions related to Hall angle.

(6) Multiple typos could be corrected now, or in the proofs: “are are”, “gaping”, “gaped”, “this is because that”.

Response: We thank the referee for carefully reading our manuscript. We have corrected all these typos.

Response to Reviewer 3

In the manuscript “Giant room temperature anomalous Hall effect and magnetically tuned topology in the ferromagnetic Weyl semimetal Co₂MnAl”, P. Li et al. have presented large anomalous Hall effect and Hall angle up to room temperature, which they attribute to the Berry curvature induced by non-trivial band topology. Then they propose a mechanism to generate large anomalous Hall effect by gapping nodal rings without requiring the existence of Weyl points. Single crystal growth has helped reveal one of the highest values of anomalous Hall effect and the Hall angle at room temperature. The theoretical calculations support the observation of the Hall effect, and this is a topical subject. I think the results presented here will be of interest to the readers in the field. However, there are few things that I would like to see explained.

Response: We thank the referee for positive assessment of our work. We also appreciate the referee’s insightful comments, which have been very helpful in improving our manuscript. We have addressed all the issues raised by the referee and revised the manuscript accordingly. The following is our point-to-point response to the referee’s comments.

1) I think separation of anomalous Hall effect due to simple ferromagnetic contribution (say if the band structure was that of a simple metal) would be helpful to better understand the actual contribution coming from the non-trivial band structure. It may also help to see if the anomalous Hall conductivity value is intrinsically related to the Berry curvature or has some contribution from the ferromagnetic ordering. It may be that the ferromagnetic Hall conductivity is small compared to the one observed here. But separating these two contributions will help readers. In other words, separating normal, anomalous (proportional to the magnetization) and topological Hall components would make it more clear.

Response: Anomalous Hall effect (AHE) in a simple ferromagnetic (FM) metal without non-trivial band topology is usually weak, and its origin is extrinsic and generally attributed to spin-dependent scattering involving skew scattering and side jump (Nagaosa et al., Rev. Mod. Phys. 82, 1539 (2010)). In contrast, for a FM material with non-trivial band topology, its intrinsic anomalous Hall response due to Berry curvature can be very strong, which is manifested by large anomalous Hall conductivity and anomalous Hall angle.

In the manuscript, we have shown Co₂MnAl features both large anomalous Hall conductivity and an exceptionally large anomalous Hall angle even at room temperature. Both our experimental and theoretical studies have demonstrated the large anomalous Hall response in Co₂MnAl is indeed of intrinsic origin. Referee 2 also asked one question relevant to this issue (i.e. question #4). In our response to his question (see above), we have shown while the longitudinal resistivity $\rho_{xx}(T)$ exhibits clear temperature dependence and localization behavior [Figure 4 (for referee)], indicating the scattering rate varies with temperature, the corresponding anomalous Hall conductivity σ_{yx} remains nearly unchanged from several K to 300 K for I//(111) (Fig.2b), which is a direct proof of the intrinsic nature of the AHE. For I//(001), although σ_{yx} exhibits a slightly negative slope as compared to the I//(111) case, this can be rationalized as temperature-dependence of the Berry

curvature due to the Fermi-Dirac distribution. At theoretical side, we have reproduced both the large values of σ_{yx} and its weak temperature dependence based on the intrinsic mechanism (see supplementary Fig. S1). Furthermore, our theoretical work reveals the Berry curvature of gapped nodal rings is the origin of intrinsic anomalous Hall response in Co₂MnAl. The quantitative consistency between our experiments and theory implies the anomalous Hall response due to extrinsic effect is negligible.

In Co₂MnAl, the normal Hall contribution due to the Lorentz effect ($\rho_{xy}^0 = R_0B$) is also tiny as compared to the anomalous contribution $\rho_{xy}^A (=4\pi R_sM)$ since ρ_{xy} shows nearly the same magnetic field dependence as the isothermal magnetization. ρ_{xy}^0 can be evaluated from its linear field dependence in the field range where the magnetization is saturated. From the data shown in Fig. 2a, we find ρ_{xy}^0 is ~ only 0.3-3% of ρ_{xy} in the 2-9T field range where the magnetization is saturated. We have subtracted this normal Hall contribution before converting it to anomalous Hall conductivity in the revised manuscript. Since ρ_{xy}^0 is very small, the new Hall conductivity and Hall angle data (Fig.2b and 2c) obtained after subtracting normal Hall contribution is almost the same as the previous data shown in the original manuscript.

Finally, it is worth noting that the topological Hall effect (THE) due to Berry curvature in magnetic systems with non-collinear spin structures (e.g. a canted antiferromagnetic state) can be separated from the anomalous Hall resistivity linearly coupled with magnetization, as demonstrated in GdPtBi (Suzuki et al., Nat. Phys. 12, 1119(2016)). This is because the Berry curvature in such systems originate from the combined effect of strong spin-orbit coupling and broken time-reversal and lattice symmetries, but not depend on magnetization (Chen et al., PRL 112, 017205(2014)). Such a picture apparently does not apply to Co₂MnAl.

In the revised manuscript, we have added additional remarks on extrinsic AHE as well as the magnitude of the normal Hall contribution. Moreover, we have also added more discussions on the intrinsic origin of AHE of Co₂MnAl.

2) *I would like to see the discussion about how different the proposed mechanism for the large Hall effect presented here is from that discussed in GdPtBi by Joe Checkelsky's group [Reference 1 in supplementary materials)?*

Response: In GdPtBi (Suzuki et al., Nat. Phys. 12, 1119(2016)), the AHE includes two contributions: the topological Hall effect from nonlinear spin structure and the Berry curvature related to the Weyl points. Although the large AHE of Co₂MnAl is also driven by an external magnetic field as in GdPtBi, its AHE does not involve a topological Hall component related with non-collinear spin structure, since Co₂MnAl exhibits only a collinear ferromagnetic state. Furthermore, we stress that the large AHE in Co₂MnAl is predominantly related to the Berry curvature of gapped nodal lines, rather than Weyl points. We pointed it out in the end of the introduction by “This mechanism is distinct from the previous understanding of the AHE originating from Weyl points for Co₂MnGa and similar materials.”.

3) Experimental data presented in Fig. 3(a) needs more details. How is the magnetization angle

changed experimentally? If it is by tilting the magnetic field, then what is the relation between current, magnetic field and voltage?

Response: In our experimental set-up, the magnetization angle was changed by tilting the magnetic field. As shown in the figure attached below, the current is applied parallel to (001) plane and the Hall voltage is measured along the direction perpendicular to the current. The magnetic field is rotated on the y-z plane, i.e. the (100) plane. The Hall conductivity data presented in Fig. 3(a) were obtained under a magnetic field of $B=2$ T. Under such a field, the magnetization is saturated, and is driven to rotate with the rotation of magnetic field. We have added the schematic of the experiment set-up to Fig. 1a as an inset.

Figure 5 (for referee 3). The schematic diagram of the sample geometry for the configuration

4) It would be helpful to see ρ_{xx} vs H and ρ_{xx} vs T . What is the RRR value of the measured crystal etc.

Response: As shown in the figure attached below [Fig.6 (for referee)], the RRR ratio $[R(300K)/R(5K)]$ of two samples for $I//(001)$ and $I//(111)$ is 1.04 and 1.02, respectively. Both samples show small negative magnetoresistance behavior, as shown in Fig. 7 attached below. We have added these data to the supplementary materials.

Figure 6 (for referee). Temperature dependence of resistivity for two samples with $I//(001)$ and $I//(111)$.

Figure 7 (for referee). : Magnetic field dependence of longitudinal resistivity R_{xx} for two samples with $I // (001)$ and (111) , respectively.

5) I prefer candidate Weyl (semi)metal in the title as opposed to Weyl semimetal at this point unless supported by other experimental results for the Weyl (semi)metallic behavior. I think showing band structure over the whole Brillouin zone would be helpful (either in the main text or even in supplementary section. Without seeing a full band structure I am not sure whether to call it a metal or a semimetal. Are there any other experimental evidence to call it a semimetal?

Response:

Following the referee's suggestion, we have replaced "Weyl semimetal" with "candidate Weyl semimetal" in the title. We have also added one band structure to the supplementary section. From the band structure, it seems more appropriate to call this material a metal, since the carrier density is high and the Fermi surface is large. We use the terminology Weyl semimetal for the conventional reason. It indeed has two types of carriers: normal electrons of large density, and topological carriers of small density. However, the anomalous Hall effect is predominantly contributed by the topological carrier. We have added these remarks to the manuscript to avoid misunderstanding.

Summary of changes

We have addressed all the issues raised by the reviewers and revised the manuscript accordingly (highlighted in blue). Here is the summary of changes made in the manuscript:

1. We have changed “ferromagnetic Weyl semimetal” to “ferromagnetic Weyl semimetal candidate” in the title.
2. We have added two insets in Fig. 1e to show the remnant magnetizations.
3. We have also added two insets to Fig. 2a to show the evidence of electronic band structure reconstruction induced by the changes of magnetization orientation. We also added a schematic to show the experimental set-up for the angular dependent Hall effect measurements. The two insets in Fig. 2a and one inset in Fig. 2b in the original manuscript have been removed. We have also updated the figure caption of Fig. 2.
4. We adopted the definition of Hall angle suggested by the referee, i.e. $\tan\theta_H = \sigma_{yx}/\sigma_{xx}$.
5. In the left column of Page 2, we have added remarks to indicate our Co_2MnAl crystals are structurally different from previously reported Co_2MnAl thin films.
6. We have added the remnant magnetization information in the right column of page 2.
7. We have added more discussions on the different contributions to Hall effect (i.e. normal Hall contribution, intrinsic anomalous Hall contribution and extrinsic anomalous Hall contribution) on Page 2 and 3.
8. We have revised the discussions of the evidence of prominent intrinsic anomalous Hall contribution on Page 4.
9. We have also added additional comments on Page 4 to show different dependence of intrinsic anomalous Hall conductivity on the chemical potential between $B//[001]$ and $B//[111]$.
10. We have added discussions on Page 6 to show the clear evidence of band structure reconstruction with the rotation of magnetization.
11. We have added the band structure in the full Brillouin zone to supplementary Fig. S4
12. We have added the data of field dependence of longitudinal resistivity to supplementary Fig. S5
13. We have added the data of the temperature dependence of resistivity to supplementary Fig. S6
14. We have added the data of the field dependence of Hall resistivity at various magnetization orientation angles to supplementary Fig. S7.

REVIEWER COMMENTS

Reviewer #1 (Remarks to the Author):

In the revised manuscript, Peigang Li et al. clarified some issues, which were not clearly stated in the previous manuscript, and also provided new experimental results, which somewhat improve the quality of the work. However, I don't think that the critical issues that I raised in the previous review are fully clarified, as elaborated below in detail.

Firstly, I do understand that the crystal structure of Co₂MnAl single crystal in this work is the L21 type, different from the B2 type of Co₂MnAl film in the literature. However, as I mentioned in the previous review, the anomalous Hall conductivity of the film is $\sim 2000 \text{ Ohm}\cdot\text{cm}^{-1}$, comparable with the maximum value $\sim 1600 \text{ Ohm}\cdot\text{cm}^{-1}$, obtained the single crystal in this work. Thus the experimental observation of the large anomalous Hall effect (AHE) in Co₂MnAl cannot guarantee the novelty of this work. Secondly, it has been well known that the avoided band crossing with spin-orbit coupling is critical to induce the AHE. Thus it quite true that the gapped nodal line contributes more to Berry curvature than the Weyl points, which cannot be considered as a new proposal.

Therefore, the novelty of this work should rely on experimental confirmation of the tunable AHE and the novel mechanism for it. As the authors emphasized in the abstract, the main claim of this work is that the transition between Weyl points and nodal rings is responsible for the tunable AHE under the rotating magnetic field. I do agree on the theoretical proposal that the opening and closing of the SOC gap depends on the magnetic field orientation, resulting in the creation or annihilation of the Weyl points. However, I think it is a separate issue whether or not such a field-induced topological change is indeed experimentally verified in this work. If the theoretical calculations predict the simple $\cos(\theta)$ dependence of Hall conductivity, it may indicate that the transition of Weyl points with rotating magnetic fields gives a negligible contribution to the AHE. As discussed in the manuscript, there are four nodal lines that contribute to Berry curvature. The one denoted #3, remains gapped during rotating magnetic fields and thus is the main source of Berry curvature. The other nodal rings hosting Weyl points with the rotating magnetic field seem not important to determine the observed angle-dependent AHE. If this is the case, I think that the proposed novel mechanism for the AHE is masked by the simple one without topological transition. The quantitative analysis about the contribution of transition between Weyl points and nodal rings would be helpful to clarify this issue.

In conclusion, I do not recommend publication of this work as it is. If the authors provide a more convincing explanation or evidence, I can reconsider my opinion on this work.

Reviewer #2 (Remarks to the Author):

I was recently asked to review a mixed theoretical and experimental paper by Li et al. on the giant room temperature anomalous Hall effect in Co₂MnAl.

My most severe criticism was that the main claim of the paper - i.e. that the Hall response can be controllably tuned by modifying the topological band structure in a magnetic field - was not supported sufficiently experimentally. As a matter of fact, Referee 1 raised similar concerns, which I will not further comment upon.

In their response, the authors carefully address all of my questions and have updated the manuscript accordingly. In regards of the main criticism, they explain why complementary experimental

techniques to corroborate their interpretation were not available. In the updated manuscript, they also included additional details on transport measurements which are consistent with the theory.

To me, this timely paper reports interesting experimental transport data and an elaborate theory for a novel, candidate topological material. While I still think that it is questionable to infer that the experimental data confirms the theory or vice versa, I do agree that they are mutually consistent. Therefore, I invite authors to tune down the language (e.g. replace "confirm" by "consistent with" etc.) in the corresponding parts of the paper, particularly in abstract and introduction. If this will occur, I am ready to suggest this paper for publication in Nature Comm.

One more concrete scientific comment: I do not understand the statement "As seen in Fig. 2a, ρ_{xy} shows nearly the same magnetic field dependence as the isothermal magnetization curve shown in Fig.1e," on p. 2 of the updated manuscript. Clearly, the magnetic field at which M and ρ_{xy} saturate is rather different (most drastically for B \perp (001)). This needs to be clarified.

Reviewer #3 (Remarks to the Author):

I still think that the results the authors have obtained both experimentally and theoretically are interesting and novel to get published in Nature communications. Especially, the contribution to the AHE due to Berry curvature by opening of gap at the nodal lines is new. There have been previous theoretical attempts (at least at the conceptual level) e.g. from Rim and Kim, PRB 92, 045126 (2015) (might worth citing) on such a system but not directly experimentally observed at least to my knowledge.

I however, suggest the authors to consider the following point regarding my previous concern regarding the effect of ferromagnetic ordering. The authors replied that ferromagnets without non-trivial band topology have negligible anomalous Hall. At least on the basis of the experimental results, I find it hard to accept. An example is Fe_{1/4}TaS₂ [PRB 77, 014433 (2008)], which is a ferromagnet, so far, no non-trivial bands are reported. But has a significant AHE and is believed to be due to an intrinsic Berry phase related effect. I agree with the fact that the contribution to the AHE due to ferromagnetic ordering in the Co₂MnAl may be small compared to what is observed experimentally. But I think the possibility of the AHE due to ferromagnetic ordering (say if there was no topological contribution) cannot be ignored, unless the authors have a reason to do so on this particular compound. This then may change the value of the conductivity the authors reported to be close to the quantized Hall value. I think the authors need to either find this contribution due to ferromagnetic ordering or if the ferromagnetic contribution is difficult to separate then acknowledge it in the write up so that readers know that there can be a ferromagnetic contribution in addition to that coming from the gap opening at the nodal lines. At the least, it is clear that Co₂MnAl has a significant anomalous Hall contribution that a simple ferromagnet cannot have.

Response to referee 1:

Comment:

In the revised manuscript, Peigang Li et al. clarified some issues, which were not clearly stated in the previous manuscript, and also provided new experimental results, which somewhat improve the quality of the work. However, I don't think that the critical issues that I raised in the previous review are fully clarified, as elaborated below in detail.

Reply:

We thank the referee for his/her review on our manuscript.

Comment:

Firstly, I do understand that the crystal structure of Co₂MnAl single crystal in this work is the L21 type, different from the B2 type of Co₂MnAl film in the literature. However, as I mentioned in the previous review, the anomalous Hall conductivity of the film is $\sim 2000 \text{ Ohm}\cdot\text{cm}^{-1}$, comparable with the maximum value $\sim 1600 \text{ Ohm}\cdot\text{cm}^{-1}$, obtained the single crystal in this work. Thus the experimental observation of the large anomalous Hall effect (AHE) in Co₂MnAl cannot guarantee the novelty of this work. Secondly, it has been well known that the avoided band crossing with spin-orbit coupling is critical to induce the AHE. Thus it quite true that the gapped nodal line contributes more to Berry curvature than the Weyl points, which cannot be considered as a new proposal.

Reply:

As the referee has been aware, the L21 and B2 phases are structurally different: Mn and Al are ordered in the L21 phase, but disordered in the B2 phase. It is the L21 phase that is predicted to have non-trivial band topology and large AHE, while the band structure and topological properties of the B2 phase remain unknown. Furthermore, the previously reported thin film samples are polycrystalline; boundary scattering must be involved in transport, which may mask intrinsic AHE. As such, comparing our results obtained in Co₂MnAl crystals with the L21 phase with that of polycrystalline sample with the B2 phase may not be appropriate.

The novelty of our experimental results includes the first demonstration of intrinsic AHE of the L21 phase of Co₂MnAl as well as the first observation of the record value of anomalous Hall angle at room temperature (Fig. 2d). In the APL (2011) paper the referee cited, the authors did not provide any signature and analysis of the intrinsic or extrinsic nature of AHE. Further, we should point out that the anomalous Hall conductivity (AHC) of $2000 \text{ }\Omega\cdot\text{cm}^{-1}$ is not a large value if it is from an extrinsic AHE. For example, super-clean Fe metal can exhibit huge AHC $\sim 10^4 \text{ -- } 10^5 \text{ }\Omega\cdot\text{cm}^{-1}$ in the extrinsic regime (see Figure 5 in PRB 79, 014431 (2009)). The extrinsic AHE is related to the scattering and cannot represent the band structure topology. Instead of the AHC, the Hall angle (AHC divided by the normal conductivity) is more meaningful to represent the strength of AHE and the topology as well. As we stressed in the abstract and in the paper, we observed a record value of the Hall angle ~ 0.21 from the intrinsic AHE.

We acknowledge the previous Berry curvature calculations by Kuebler and Felser et al predicted large AHE for the L21 phase of Co₂MnAl and we have cited those works. However, we hope that the referee would recognize our new theoretical finding of a fully gapped nodal ring that can generate a large intrinsic AHE. Conceptually, this mechanism is distinguished from the AHE mechanism of the band anti-crossing and Weyl points. The nodal ring scenario provides new physics underlying the band structure and AHE. It represents a nearly quantized AHE (not fully quantized because of coexisting trivial bands), with Hall conductance of $2e^2/h$, as stressed in our work. This insightful estimation is very close to the numerical value. As far as we are aware, there has been no previous theoretical and experimental report, which includes studies on Co₂MnAl and all other AHE materials, to appreciate the contribution of the fully gapped nodal ring to the nearly quantized AHC.

Comment:

Therefore, the novelty of this work should rely on experimental confirmation of the tunable AHE and the novel mechanism for it. As the authors emphasized in the abstract, the main claim of this work is that the transition between Weyl points and nodal rings is responsible for the tunable AHE under the rotating magnetic field. I do agree on the theoretical proposal that the opening and closing of the SOC gap depends on the magnetic field orientation, resulting in the creation or annihilation of the Weyl points. However, I think it is a separate issue whether or not such a field-induced topological change is indeed experimentally verified in this work. If the theoretical calculations predict the simple $\cos(\theta)$ dependence of Hall conductivity, it may indicate that the transition of Weyl points with rotating magnetic fields gives a negligible contribution to the AHE. As discussed in the manuscript, there are four nodal lines that contribute to Berry curvature. The one denoted #3, remains gapped during rotating magnetic fields and thus is the main source of Berry curvature. The other nodal rings hosting Weyl points with the rotating magnetic field seem not important to determine the observed angle-dependent AHE. If this is the case, I think that the proposed novel mechanism for the AHE is masked by the simple one without topological transition. The quantitative analysis about the contribution of transition between Weyl points and nodal rings would be helpful to clarify this issue.

Reply:

As we mentioned above, the AHE is dominantly contributed by the fully gapped nodal ring. Therefore, we stress in the abstract “to generate the giant anomalous Hall effect by gapping nodal rings without requiring the existence of Weyl points”. This is the mechanism distinguished from previous work, which regarded AHE rooting in the Weyl points or anti-crossing gap. It is a typical topological transition for creating and removing of nodal rings by rotating the magnetic field. In addition, the nodal ring – Weyl point transition exists, but contribute much less to AHE compare to the fully gapped rings.

We felt there is one sentence in the abstract which probably misled the referee - “We observe a tunable, giant anomalous Hall effect which is induced by the transition between Weyl points and nodal rings as rotating the magnetization axis”. In the new version, we modify it to “We observe a tunable, giant anomalous Hall effect, which is induced by the transition involving Weyl points and nodal rings as rotating the magnetization axis.”

Comment:

In conclusion, I do not recommend publication of this work as it is. If the authors provide a more convincing explanation or evidence, I can reconsider my opinion on this work.

Reply:

After we clarify several conclusions and their differences from previous works, we truly hope that the Referee can recognize their significance and reconsider his/her opinions on our work.

Response to referee 2

I was recently asked to review a mixed theoretical and experimental paper by Li et al. on the giant room temperature anomalous Hall effect in Co₂MnAl.

My most severe criticism was that the main claim of the paper - i.e. that the Hall response can be controllably tuned by modifying the topological band structure in a magnetic field - was not supported sufficiently experimentally. As a matter of fact, Referee 1 raised similar concerns, which I will not further comment upon.

In their response, the authors carefully address all of my questions and have updated the manuscript accordingly. In regards of the main criticism, they explain why complementary experimental techniques to corroborate their interpretation were not available. In the updated manuscript, they also included additional details on transport measurements which are consistent with the theory.

To me, this timely paper reports interesting experimental transport data and an elaborate theory for a novel, candidate topological material. While I still think that it is questionable to infer that the experimental data confirms the theory or vice versa, I do agree that they are mutually consistent. Therefore, I invite authors to tune down the language (e.g. replace “confirm” by “consistent with” etc.) in the corresponding parts of the paper, particularly in abstract and introduction. If this will occur, I am ready to suggest this paper for publication in Nature Comm.

Reply: We thank the referee for taking time again to review our manuscript and giving positive comments and constructive suggestions. Following the referee’s suggestion, we have tuned down the language of the manuscript and avoided using “confirm” or “demonstrate” when describing the comparison between theory and experiment. Instead, we stressed the consistency between theory and experiment.

One more concrete scientific comment: I do not understand the statement “As seen in Fig. 2a, ρ_{xy} shows nearly the same magnetic field dependence as the isothermal magnetization curve shown in Fig. 1e,” on p. 2 of the updated manuscript. Clearly, the magnetic field at which M and ρ_{xy} saturate is rather different (most drastically for B \perp (001)). This needs to be clarified.

Reply: We really appreciate that the referee found this problem. The field orientation in the upper panel of Fig. 1e is mislabeled, i.e, B// (001) and B \perp (001) should be swapped. We apologize for making this mistake. The Hall resistivity and isothermal magnetization indeed follow a similar field dependence as shown in the figure attached below. We have corrected this mistake in the revised manuscript. The statement “As seen in Fig. 2a, ρ_{xy} shows nearly the same magnetic field dependence as the isothermal magnetization curve shown in Fig. 1e,” has also been changed to “As seen in Fig. 2a, ρ_{xy} shows a similar magnetic-field dependence as the isothermal magnetization shown in Fig. 1e,”

Response to referee 3

I still think that the results the authors have obtained both experimentally and theoretically are interesting and novel to get published in Nature communications. Especially, the contribution to the AHE due to Berry curvature by opening of gap at the nodal lines is new. There have been previous theoretical attempts (at least at the conceptual level) e.g. from Rim and Kim, PRB 92, 045126 (2015) (might worth citing) on such a system but not directly experimentally observed at least to my knowledge.

Reply: We thank the referee for reviewing our manuscript and seeing the novelty of our work. We also appreciate the referee pointing out the reference and find it very relevant. We have cited their paper in the revised manuscript.

I however, suggest the authors to consider the following point regarding my previous concern regarding the effect of ferromagnetic ordering. The authors replied that ferromagnets without non-trivial band topology have negligible anomalous Hall. At least on the basis of the experimental results, I find it hard to accept. An example is $\text{Fe}_{1/4}\text{TaS}_2$ [PRB 77, 014433 (2008)], which is a ferromagnet, so far, no non-trivial bands are reported. But has a significant AHE and is believed to be due to an intrinsic Berry phase related effect. I agree with the fact that the contribution to the AHE due to ferromagnetic ordering in the Co_2MnAl may be small compared to what is observed experimentally. But I think the possibility of the AHE due to ferromagnetic ordering (say if there was no topological contribution) cannot be ignored, unless the authors have a reason to do so on this particular compound. This then may change the value of the conductivity the authors reported to be close to the quantized Hall value. I think the authors need to either find this contribution due to ferromagnetic ordering or if the ferromagnetic contribution is difficult to separate then acknowledge it in the write up so that readers know that there can be a ferromagnetic contribution in addition to that coming from the gap opening at the nodal lines. At the least, it is clear that Co_2MnAl has a significant anomalous Hall contribution that a simple ferromagnet cannot have.

Reply: We thank the referee for stressing this relevant question. We recall that the intrinsic AHE, by definition, is determined by the material band structure, rather than extrinsic effects like scattering by phonons, defects or disordering. If a material is FM, its band structure exhibits the spin splitting. When we estimate the AHE by the band Berry phase, we equivalently consider the FM contribution via the spin splitting in the band structure. In other words, the band structure and the FM contributions are equivalent to the AHE. For example, let's consider the "band structure of a simple metal" (asked by the referee in the 1st round review), like Cu. Its band structure exhibits no spin splitting and zero net Berry curvature because the time-reversal symmetry. Then we introduce FM and consider a FM metal for example the Fe (Let's consider only the intrinsic AHE in Fe). Actually, Fe band structure exhibits strong spin-splitting and also strong Berry curvature. Band structure calculations revealed contributions of band anti-crossings [Phys. Rev. Lett. 92, 037204 (2004)]. After 11 years, Fe was even reported to exhibit Weyl points in the band structure [Phys. Rev. B 92, 085138 (2015)].

The AHE of $\text{Fe}_{1/4}\text{TaS}_2$ is a very interesting. The abstract [PRB 77, 014433 (2008)] stated "At low temperature T (5 – 50 K), AHC is T -independent, consistent with the Berry-phase / Karplus-Luttinger theory. Above 50 K, we extract an inelastic AHE conductivity ..." They claimed the intrinsic AHC is due to the Berry-phase. This is also the case of our material. Thus, we do not have to double-count the FM contribution to the AHC.

Possibly the term “non-trivial band structure” that we used is misleading. For “non-trivial”, we meant the general features that contribute the Berry curvature, which includes the band anti-crossing, Weyl points and nodal rings etc.

We thank the referee for point out the interesting material $\text{Fe}_{1/4}\text{TaS}_2$. Indeed, there is no band topology and AHE calculations for $\text{Fe}_{1/4}\text{TaS}_2$. A recent paper [PRB 96, 205119 (20170)] reported the spin polarized band structure (see Figure 3b there), as copied in the following. Two spin channels cross each other near the Fermi energy. If introducing SOC, which is necessary for the spinful band structure, we expect that these crossing points will possibly evolve into anti-crossing gaps or Weyl points. The material surely deserves further investigation in the future work.